# Pregnancies and Neonatal Outcomes in Patients with Sickle Cell Disease (SCD): Still a (High-)Risk Constellation?

**DOI:** 10.3390/jpm11090870

**Published:** 2021-08-30

**Authors:** Pia Proske, Laura Distelmaier, Carmen Aramayo-Singelmann, Nikolaos Koliastas, Antonella Iannaccone, Maria Papathanasiou, Christian Temme, Hannes Klump, Veronika Lenz, Michael Koldehoff, Alexander Carpinteiro, Hans Christian Reinhardt, Angela Köninger, Alexander Röth, Raina Yamamoto, Ulrich Dührsen, Ferras Alashkar

**Affiliations:** 1Department of Hematology and Stem Cell Transplantation, West German Cancer Center, University Hospital Essen, University of Duisburg-Essen, 45147 Essen, Germany; pia.proske@uk-essen.de (P.P.); laura.distelmaier@vivantes.de (L.D.); michael.koldehoff@uk-essen.de (M.K.); alexander.carpinteiro@uk-essen.de (A.C.); christian.reinhardt@uk-essen.de (H.C.R.); alexander.roeth@uk-essen.de (A.R.); ulrich.duehrsen@uk-essen.de (U.D.); 2Vivantes, MVZ Neukölln, 12351 Berlin, Germany; 3Department of Pediatrics III, University Children’s Hospital Essen, University of Duisburg-Essen, 45147 Essen, Germany; carmen.aramayo-singelmann@uk-essen.de; 4Department of Gynecology and Obstetrics, University of Duisburg-Essen, 45147 Essen, Germany; nikolaos.koliastas@uk-essen.de (N.K.); antonella.iannaccone@uk-essen.de (A.I.); angela.koeninger@barmherzige-regensburg.de (A.K.); 5Department of Cardiology and Vascular Medicine, West German Heart and Vascular Center, Medical Faculty, University Hospital Essen, 45147 Essen, Germany; maria.papathanasiou@uk-essen.de; 6Institute for Transfusion Medicine, University Hospital Essen, University of Duisburg-Essen, 45147 Essen, Germany; christian.temme@uk-essen.de (C.T.); hannes.klump@uk-essen.de (H.K.); veronika.lenz@uk-essen.de (V.L.); 7Institute for Molecular Biology, University of Duisburg-Essen, 45147 Essen, Germany; 8Hospital of the Order of St. John of God Regensburg, Clinic for Gynaecology and Obstetrics, 93049 Regensburg, Germany; 9MVZ Dr. Eberhard & Partner, 44137 Dortmund, Germany; yamamoto@labmed.de

**Keywords:** pregnancy, sickle cell disease, vaso-occlusive (VOC), transfusion, complications, genotype–phenotype correlation and patient stratification, patient registries and standardization

## Abstract

Background: This monocentric study conducted at the University Hospital of Essen aims to describe maternal and fetal/neonatal outcomes in sickle cell disease (SCD) documented between 1996 to 2021 (N = 53), reflecting the largest monocentric analysis carried out in Germany. Methods/Results: 46 pregnancies in 22 patients were followed. None of the patients died. In total, 35% (11/31) of pregnancies were preterm. 15 pregnancies in eight patients were conceived on hydroxycarbamide (HC), of which nine had a successful outcome and three were terminated prematurely. There was no difference regarding the rate of spontaneous abortions in patients receiving HC compared to HC-naive patients prior to conception. In patients other than HbS/C disease, pregnancies were complicated by vaso-occlusive crises (VOCs)/acute pain crises (APCs) (96%, 23/24); acute chest syndrome (ACS) (13%, 3/24), transfusion demand (79%, 19/24), urinary tract infections (UTIs) (42%, 10/24) and thromboembolic events (8%, 2/24). In HbS/C patients complications included: VOCs/APCs (43%, 3/7; ACS: 14%, 1/7), transfusion demand (14%, 1/7), and UTIs (14%, 1/7). Independent of preterm deliveries, a significant difference with respect to neonatal growth in favor of neonates from HbS/C mothers was observed. Conclusion: Our data support the results of previous studies, highlighting the high rate of maternal and fetal/neonatal complications in pregnant SCD patients.

## 1. Introduction

Over the last years, sickle cell disease (SCD)-related maternal and neonatal mortality has decreased significantly. This can be ascribed to the availability of evidence-based clinical practice guidelines (CPGs) and preventive measures, antenatal counseling and implementation of newborn screening [1,2]. However, pregnancy in patients with SCD still remains associated with both maternal morbidity and fetal/neonatal morbidity and mortality [3,4]. The observation of increased maternal morbidity is supported by the fact that care in SCD throughout pregnancy and in the postpartum period is mainly restricted to supportive measures in addition to transfusion therapy. The reason is that therapeutic modalities require immediate discontinuation due to the fear of possible complications that could particularly endanger fetal well-being [5,6,7]. Therefore, it is not surprising that sickle cell-associated complications (e.g., anemia, vaso-occlusive crises (VOCs), including acute chest syndrome (ACS), pre-existing cardio-pulmonary and renal complications) often deteriorate in pregnant SCD patients, resulting in frequent inpatient admissions for medical care. In addition, infectious complications, thromboembolic events (TEs), and preeclampsia can further complicate pregnancy and the postpartum period [8].

Erythrocyte sickling may contribute to micro-vascular placental pathology, increasing the risk for decreased placental circulatory blood flow, (acute) fetal hypoxemia, and placental thromboses, all of which are closely associated with a high incidence of perinatal complications. Those can be spontaneous abortions, stillbirths, or an increased rate of preterm deliveries with low gestational weight in neonates. Often, there is an imperative need for cesarean deliveries [8].

The prevalence of SCD and the number of affected children born in Germany is presently unknown. Notwithstanding that reliable epidemiological data are lacking, it is estimated that approximately 3000 to 5000 affected children and adults currently live in Germany. According to conservative estimates, approximately 70 to 150 affected children are born in Germany each year. Furthermore, due to increased immigration within the past years, more children affected by SCD will be born within the next years [9].

Here we report pregnancy outcomes in patients with SCD registered at the Department of Hematology and Stem Cell Transplantation, University Hospital Essen, Germany. To the best of our knowledge, this is the first large German monocentric observational study addressing this clinically important topic in a considerable number of patients.

## 2. Materials and Methods

### 2.1. Study Design and Participants

This is a single-center observational study of pregnancies in women with homozygous SCD (HbSS) or compound heterozygous states, such as HbS/C, HbS/β-thalassemia, or HbS/O-Arab (±co-inheritance of heterozygous α-thalassemia, if tested) who were in part or entirely monitored at the Department of Hematology and Stem Cell Transplantation, University Hospital Essen, Germany, allowing a prospective follow-up. Pregnancies before referral to our department were recorded retrospectively, based on patients’ information and medical reports. Data from patients delivering outside of Germany were generally not available. Carrier states for SCD were excluded. The study was approved by the Ethics Committee of the University of Duisburg-Essen and conducted in accordance with the Declaration of Helsinki.

### 2.2. Disease-Related Definitions, Methods, and Treatments

SCD was diagnosed according to international standards at the Hemoglobin Laboratory of the University Hospital Ulm and starting from 2017 at the Medical Care Center Dr. Eberhard and Partner Dortmund via molecular globin gene genetic analyses by polymerase chain reaction (PCR), sequencing and/or multiplex ligation-dependent probe amplification (MLPA).

### 2.3. Sickle Cell-Associated Definitions

An acute pain crisis (APC) or uncomplicated VOC was defined as an acute onset of pain for which there was no other medical explanation other than vaso-occlusion and which required analgetic treatment (enteral or parenteral), irrespective if the event required hospitalization or was managed at home. ACS or splenic sequestration were referred to as complicated VOCs. ACS was defined based on the finding of a new pulmonary infiltrate in the presence of other signs and symptoms: chest pain, a temperature of ≥38.5 °C, tachypnea, wheezing or cough. Splenic sequestration was defined on the basis of left upper quadrant pain, an enlarged spleen, and an acute decrease in hemoglobin (Hb) concentration (e.g., a decrease in Hb of 2 g/dL from baseline).

### 2.4. Pregnancy-Related Definitions

Spontaneous abortion and stillbirth were defined as unsuccessful pregnancy outcomes before or after gestational week (GW) 22. Preterm delivery was defined as delivery before GW 37. Preterm premature rupture of membranes (PPROM) was classified if PROM occurred before GW 37.

### 2.5. Treatments

Treatment with hydroxycarbamide (HC) was recommended according to international guidelines and prescribed, if desired. In all HC-treated patients, therapy was discontinued with establishment of pregnancy due to possible teratogenicity [10,11]. Prior to pregnancy, none of the patients were treated with crizanlizumab. In patients allowing for pre-conceptional care (PCC), HC was aimed to be discontinued at least three months before conception under close follow-up, if feasible.

Folic acid (5 mg daily) was routinely supplemented. Iron supplementation was restricted to patients with iron deficiency. Extended red blood cell (RBC) phenotyping, including full Rhesus (C, D, E, c, and e), Kell typing (K, k) and screening for other blood group antigens [Kp(a), Kp(b), Duffy (Fy(a), Fy(b)), Kidd (Jk(a), Jk(b)), M, N, S, s, Lewis (Le(a), Le(b)), Lu(a), Lu(b)] was routinely performed. Partners of patients were encouraged to undertake genetic counseling.

In patients with past pregnancies not yet treated at our center, extended RBC phenotyping was performed on the date of their first visit. RBC transfusions were restricted to patients with symptomatic anemia, a peri- or antenatal Hb concentration <7 g/dL, or in pregnant patients with SCD- or fetal-related complications. A RBC exchange (RCE) was recommended for the treatment of severe VOCs (i.e., severe pain crises and/or ACS). Low-molecular-weight-heparins (LMWHs) were prescribed to patients with prior TEs, in the event of inpatient admission, and were routinely applied in the postpartum period. The duration of thromboprophylaxis was at the discretion of the treating physician, based on the individual risk profile and mode of delivery.

### 2.6. Statistical Analysis

Due to the small sample size, statistical analysis was not carried out except for maternal and neonatal outcomes using paired Student’s *t*-test and two-tailed Mann-Whitney U test. Differences between groups were considered to be significant at a *p* value of < 0.05. Statistical analyses were performed with GraphPad Prism 6.

## 3. Results

In total, 53 pregnancies in 22 females were recorded (HbSS (15/22; 68%)), HbS/β-thalassemia (2/22; 9%), HbS/C (4/22; 18%), and HbS/O-Arab (1/22; 5%) ± co-inheritance of heterozygous α-thalassemia) (Table 1). Prior to first presentation, seven gravidities occurred outside Germany, of which five were first-time pregnancies. The median age at first conception was 24 years (range, 14–39; N = 22). Four patients had one, six had two, and eleven had ≥3 pregnancies. No twin pregnancies were observed. In two patients the diagnosis of SCD was established after first gestation (Pat. 2 and 10).

Fifteen conceptions occurred in eight of the 22 patients under HC treatment (median maximum HC dosage: 20 mg/kg body weight (BW) (range, 8–35)), with three documented pregnancies in three patients and two in one patient. Antenatal counseling allowing for adequate time of HC discontinuation was possible in only one patient (Pat. 8).

At first presentation, alloimmunization to RBC antigens was observed in two patients (Pat. 1: anti-K and 16: anti-M, and -E). Both of which had a medical history of repeated RBC transfusions.

### 3.1. Maternal, Fetal, and Neonatal Outcomes

None of the patients died. In total, 31 of the 53 (58%) documented pregnancies in 22 patients had a successful outcome (median age at conception: 26 years, range 14–39) with 26 deliveries observed in Germany (Table 2). Excluding elective (N = 9) or medical-induced abortions (N = 2), 74% (31/42) of pregnancies were successful. Ten pregnancies in eight patients resulted in spontaneous abortions, including one patient requiring in-vitro fertilization (IVF) (28 years of age (median), range 20–33).

Overall, patient age was independent with respect to pregnancy outcome (successful versus unsuccessful), irrespective (prior) HC exposure: overall: *p =* 0.7477; severe genotypes: *p =* 0.4697. Preterm deliveries (35% (11/31)) were secondary to premature rupture of membranes (PROM), HELLP syndrome (hemolysis, elevated liver enzymes, low platelets), placenta insufficiency or abruption, pathologic fetal cardiotocographs (CTGs) and doppler indices, amniotic infection syndrome (AIS), maternal urosepsis, fetal subdural hematoma, spontaneous onset of labor in a multigravida patient, or due to a combination of two causes. Medical data concerning the reasons for preterm deliveries in patients giving birth outside Germany were in general not available.

In HC-pre-treated patients, nine of the 15 documented pregnancies had a successful outcome. Three pregnancies were terminated prematurely by elective abortions and three pregnancies resulted in spontaneous abortions, compared to six spontaneous abortions of the 22 pregnancies in patients with no prior exposure to HC. None of the patients with HbS/C disease experienced spontaneous abortions/stillbirths or received HC prior to pregnancy (median age at conception: 28 years, range 17–35).

The rate of spontaneous abortions in patients with severe genotypes, if excluding the one patient requiring IVF (independent risk factor), was independent of prior HC treatment (HC-treated: 25% (3/12); HC-naive: 27% (6/22)). Of note, two pregnancies in one patient were conceived under HC (Pat. 16).

The histological evaluation of the placenta from the one patient who experienced a stillbirth was not available. In patients suffering from recurrent (≥2) miscarriages (N = 3), only one (Pat. 19) was tested for other causes of loss of pregnancy. In another patient (Pat. 11), the diagnosis of delta-storage pool disease was confirmed during her fourth pregnancy.

### 3.2. Gestational-, Peri-, and Postpartum Phase in Patients with Severe Genotypes (HbSS, HbS/β-Thalassemia, and HbS/O-Arab Disease)

In patients with severe genotypes and successful pregnancies (N = 24), APCs or VOCs were described by all patients during gestation, except for one carrying the HbSS genotype (96%, 23/24). These events predominately occurred during the second and third trimester, requiring in part (recurrent) inpatient admissions (54%, (13/24)) for (parenteral) analgetic treatment, and/or RBC transfusion support secondary to VOCs, or because of (infectious-related) hemolytic and/or symptomatic anemia. Throughout the observation time, two complicated VOCs (ACS) were observed in one HC-naive HbSS patient (Pat. 4) during her two successive pregnancies (GW 15 and 29 + 3) necessitating manual RCE. An additional ACS was observed in the postpartum period in a second HbSS patient (Pat. 11) (ACS rate in patients with severe disease-associated genotypes: 13% (3/24)).

RBC transfusions were necessary in 75% (18/24) of the patients. One HC-naive patient treated in Italy presenting with recurrent VOCs (including ACS) prior to pregnancy, received serial prophylactic (exchange) blood transfusions (SP(E)BTs) for four years which were continued throughout pregnancy and thereafter every three weeks (Pat. 15). In another pre-conceptionally HC-treated patient (Pat. 8), SPBTs (every three weeks, starting from GW 20) were indicated due to suspicious doppler indices and symptomatic/hemolytic anemia.

One possible delayed hemolytic transfusion reaction (DHTR) with diagnosis of an alloantibody against the S-antigen (DD hyper-hemolytic VOC) despite transfusion of extended-phenotyped and matched RBCs was observed in one HbSS patient (Pat. 16, GW 29) already expressing irregular antibodies against erythrocytic antigens (anti-E, anti-M). To avoid further RBC transfusions, resumption of HC (20 mg/kg BW) in combination with erythropoietin (EPO) in GW 34 prior to delivery in GW 35 + 2 was indicated. Due to transfusion-dependent anemia in her subsequent pregnancy despite continuation of EPO, resumption of HC (15 mg/kg BW; GW 30) was mandatory prior to delivery (GW 34 + 4). In the presence of HC, breastfeeding was avoided in this patient.

Overall, thromboembolic complications were seen in two patients (left DVT (Pat. 19; GW 26) and pulmonary embolism in the context of ACS during puerperium (Pat. 11)). Infectious complications were restricted to (recurrent) urinary tract infections (UTIs) and documented in ten out of the 24 successful pregnancies (42%), including maternal urosepsis in third trimester (GW 28) in one patient (Pat. 11).

The peripartum complications included P(P)ROM in five pregnancies, possibly associated with or occurred independently to placental abruption, cervical insufficiency, or amniotic infection syndrome (AIS). One patient suffered from HELLP syndrome. VOCs/APCs and/or RBC transfusion demand in the peripartum period were observed in five of the pregnancies (21%). A caesarean section (CS) secondary to pregnancy-, including active-phase arrest and/or pathologic doppler indices (N = 4), or SCD-related complications was mandatory in 42% (10/24) of patients for whom medical data were available, resulting in an overall CS rate of 67% (16/24). In the postpartum period, RBC transfusions were required in four patients. Postpartum cardiomyopathy was seen in one patient (Pat. 11).

Following delivery, HC was re-initiated, if breastfeeding was not desired. In breasfeeding patients, treatment was re-initiated as soon as weaning was ensured.

### 3.3. Gestational-, Peri-, and Postpartum Phase in Patients with HbS/C Disease

In the four patients with HbS/C disease, recurrent VOCs/APCs were observed in two patients. Both patients had a medical history of VOCs/APCs, however, none of them resulted in inpatient admission during pregnancy. One of the pregnancies was further complicated by transfusion demand for RBCs due to symptomatic anemia unrelated to a VOC/APC. Peripartum (SCD-associated) complications were observed in one of the pregnancies and required inpatient admission due to a VOC (pyelonephritis) in the third trimester of pregnancy which progressed to an ACS, requiring CS. AIS was seen in one of the four patients. In the postpartum period, only in one of the two patients with past VOCs/APCs during gestational phase were VOCs/APCs also documented, respectively. Throughout observation time, no serious infectious- or TE complications were observed.

Three of the overall seven successful pregnancies in the other two HbS/C patients were uneventful (43%, 3/7), despite documented VOCs/APCs prior to conception in one of them.

### 3.4. Neonatal Outcomes

In patients with severe genotypes, median GW at delivery was 37 weeks (range 29–40; N = 24) compared to 38 weeks (range, 37–40; N = 6/7) in HbS/C patients. Preterm neonates defined as <37 weeks of completed gestational age, were observed in 42% (10/24) of deliveries in patients with severe genotypes. In comparison, the rate of preterm neonates from mothers with HbS/C disease for whom gestational age was available was 14% (1/7). Median overall birth weight of neonates of mothers with severe disease-associated genotypes was 2570 g (range, 946–3090 g; N = 19) (median (overall) height: 47 cm (range, 36–50; N = 17)); median (overall) head circumference: 32 cm (range, 27.2–34.5; N = 17).

Excluding preterm neonates (N = 8), data of neonatal outcomes were available from 12 neonates of HbSS patients. Median birth weight in these neonates was 2677 g (range, 1980–2090); height (median): 47.5 cm (range, 46–50); and head circumference (median): 32 cm (range, 30–34.5; N = 21)). Neonates from mothers with HbS/C disease born >37 GW displayed a median birth weight of 3370 g (range, 2890–4330), a height (median) of 55 cm (range, 51–60); and head circumference (median) of 35 cm (range, 34–36.5). Thus, there were statistically significant differences with respect to neonatal growth in favor of neonates from HbS/C mothers (Figure 1). A preterm CS (GW 29 + 5) in one patient with HbSS-α^+^-thalassemia and platelet delta-storage pool disease was required in the context of right-sided subdural fetal hematoma (1.5 × 7 cm) with midline shift and ventricular enlargement of the lateral ventricles.

## 4. Discussion

Pregnancy in SCD is associated with a high incidence of adverse events as highlighted by past meta-analyses implicating the importance of an interdisciplinary therapeutic approach by national reference centers to ensure maternal and fetal/neonatal well-being [4,12]. The results of our single-center study are in partial agreement with previous studies.

Before commenting on the findings in our patients, we have to point out on the low number of patients with severe genotypes receiving HC, given the overall benefit regarding reduction in morbidity and mortality rates in HC-treated SCD patients [13,14,15]. The low number was based on the following notion: some of the patients rejected treatment because of individual concerns. Others were not taken care by a medical reference center prior to pregnancy and, thus, had only very limited access to antenatal care. These patients presented at our department at a late stage of pregnancy which may be attributed to their refugee status, often requiring direct hospitalization due to SCD-associated complications. Furthermore, despite the rising prevalence of SCD in Germany due to recent immigration, SCD is still considered a rare disease compared to other countries. Nevertheless, as the present analysis summarizes outcomes over a period of two decades, our results support the findings of other studies stating an overall improved maternal and neonatal outcome over the past decades [3,16,17].

In a meta-analysis published by Oteng-Ntim et al. (2015), the authors estimated a sixfold increased risk of maternal mortality in HbSS females compared to non-affected persons [4]. However, mortality rates vary depending on the particular analyses, the time of data collection, and may likely be associated with a poorer health care system as compared to Europe. Even though, no increased maternal mortality is observed in developing countries with substantial experience, as published by Babah et al. (2019) [18].

Although a low prevalence of HC-induced teratogenicity can be assumed following exposure during pregnancy, treatment is supposed to be withheld three months before conception, if feasible [19]. Of note, in all our documented pregnancies (except one) with prior maternal HC exposure (treatment duration: ≥1 year), treatment was stopped only after diagnosis of pregnancy within the 1st trimester, indicating that all fetuses were exposed to HC during organogenesis (3rd to 8th week after conception). In these patients no obvious difference with regard to fetal mortality was observed (i.e., spontaneous abortions compared to HC-naive mothers) and neonatal development was not associated with any adverse events throughout observation time. Furthermore, in one of our patients with known alloantibodies and severe transfusion-dependent anemia, HC was needed to be reinitiated in the 3rd trimester of pregnancy up to four weeks prior to delivery during her two consecutive pregnancies resulting in no adverse fetal outcome. Our observations are therefore in line with the results observed in the European non-interventional, multicentric, prospective Escort-HU study. Hence, in some patients not suitable for SP(E)BT, HC (±EPO) may remain the only therapeutic option to ensure maternal and fetal well-being. Noteworthy, HC might even be considered safe in some patients if treated throughout the entire pregnancy. However, decision should be based on a case-by-case discussion and cannot be generally recommended due to the potential harmful effects [20].

Whether or not SPBTs are beneficial in pregnant SCD patients remains a matter of debate [21,22,23,24]. Importantly, transfusion techniques and timing of SP(E)BT vary throughout the different studies probably explaining the contradictory results. Our study was not designed to answer this important topic. However, we did observe a high demand for RBC transfusions, especially in HbSS patients. Taking the risk of alloimmunization into account, transfusions were restricted to patients presenting either with symptomatic/hemolytic anemia or to those with other SCD-associated or obstetric complications.

Alloimmunization is a multifactorial process and presents one of the most clinically relevant complications in the context of pregnancy. The incidence of alloimmunization is highest in SCD patients, with a reported rate of up to 47% [25]. Most of the alloantibodies are formed against Rhesus- (Rh-) and Kell- (K-) antigens [1,26]. It is assumed that this is due to the much larger variety of Rh-antigens found outside of Europe, particularly Africa, which differ from the common variants of European blood donors [1]. We observed alloimmunization in two out of eight HbSS patients with a history of periodic blood transfusions. Our low rate may be related to extended red cell antigen typing which may lower the rate of alloimmunization, although this has not yet been clearly proven [27]. Nevertheless, even in patients with a history of low exposure to allogeneic RBCs, alloimmunization can still occur at a relatively high rate, as reported for Ugandan sickle cell patients (with 6.1% alloimmunization) [28]. Although extended RBC matching has not been formally proven to contribute to a reduction of alloimmunization rates, it can speed up identification of the specificity of alloantibodies in the case of a positive antibody screen and also helps to select compatible RBCs [27]. Furthermore, some clinically significant alloantibodies can be difficult to detect because they evanesce over time and drop below the detection threshold in the immunohematology laboratory [29]. This potentiates the risk for severe transfusion-associated complications after antigen-re-exposure by transfusion of antigen-positive red blood cells. Such a booster can particularly lead to DHTRs [30]. It also emphasizes the importance of extended RBC antigen characterization (serologically and/or genetically) of donors and recipients, especially in European countries with a low disease prevalence to decrease the risk of potential transfusion reactions [31]. Particularly intrauterine transfusions can efficiently trigger alloimmunization against RBC antigens and against HLA molecules. As a matter of fact, alloimmunization against additional RBC antigens has been reported to occur in up to 25% of women already expressing an alloantibody. In contrast to transfusions in adults, extended antigen matching of donor cells (including Duffy-, Kidd-, and S-antigens) prior to intrauterine transfusion was shown to decrease the formation of alloantibodies in the mothers by 60% [32,33].

In summary, pregnant mothers suffering from SCD who have experienced severe SCD-related complications, either preceding the current pregnancy or during a previous pregnancy, may be considered for SP(E)BTs. This is the case if a high-risk exists for both, the mother and the fetus (e.g., twin pregnancies), or in patients encountering SCD-associated complications during the current pregnancy, in patients receiving HC before the gravidity, or in patients already receiving a chronic transfusion program [27]. A switch to SP(E)BTs may be worth to be considered in patients receiving crizanlizumab prior to or in the context of pregnancy for prevention of recurrent VOCs. Whether outcome might be favored by early initiation of SPEBT as described by Vianello et al. (2018) may be answered by the TAPS-2 study, as placental damage in SCD might even occur early in pregnancy [34,35].

Because not all documented pregnancies were monitored at our department, we included a patient survey. Pregnancies in our patients with successful outcomes were accompanied by a high rate of vaso-occlusive events (VOEs), especially in HbSS patients (96%) compared to HbS/C disease, requiring in part, if severe, recurrent inpatient admissions for medical care. Of note, the patient query revealed that not all of them seeked advice of their treating physicians during a painful crisis. This may offer an explanation for the overall higher VOC/APC rates described in our patients unlike other studies. Nevertheless, routine counselling was associated with a sense of security according to the patient survey, as these patients could directly contact their attending physician in the event of impending complications, if desired.

Complicated VOEs (ACS) requiring inpatient admission for partial RCE and intravenous anti-infective treatment were observed for four pregnancies (13%, 4/31), including a patient with HbS/C disease. Thus, patients with HbS/C disease also are at increased risk for serious complications when being pregnant or in the postpartum period. Nevertheless, a higher rate of complications was seen in patients with genotypes mediating severe disease.

Special attention should also be paid to the high number of elective abortions in our study. Reasons for medical abortions were related to age at time of conception, socio-economic factors, including uncertain partnerships, or disease-related stressors, implicating the importance of early psycho-social support in patients with SCD.

## Figures and Tables

**Figure 1 jpm-11-00870-f001:**
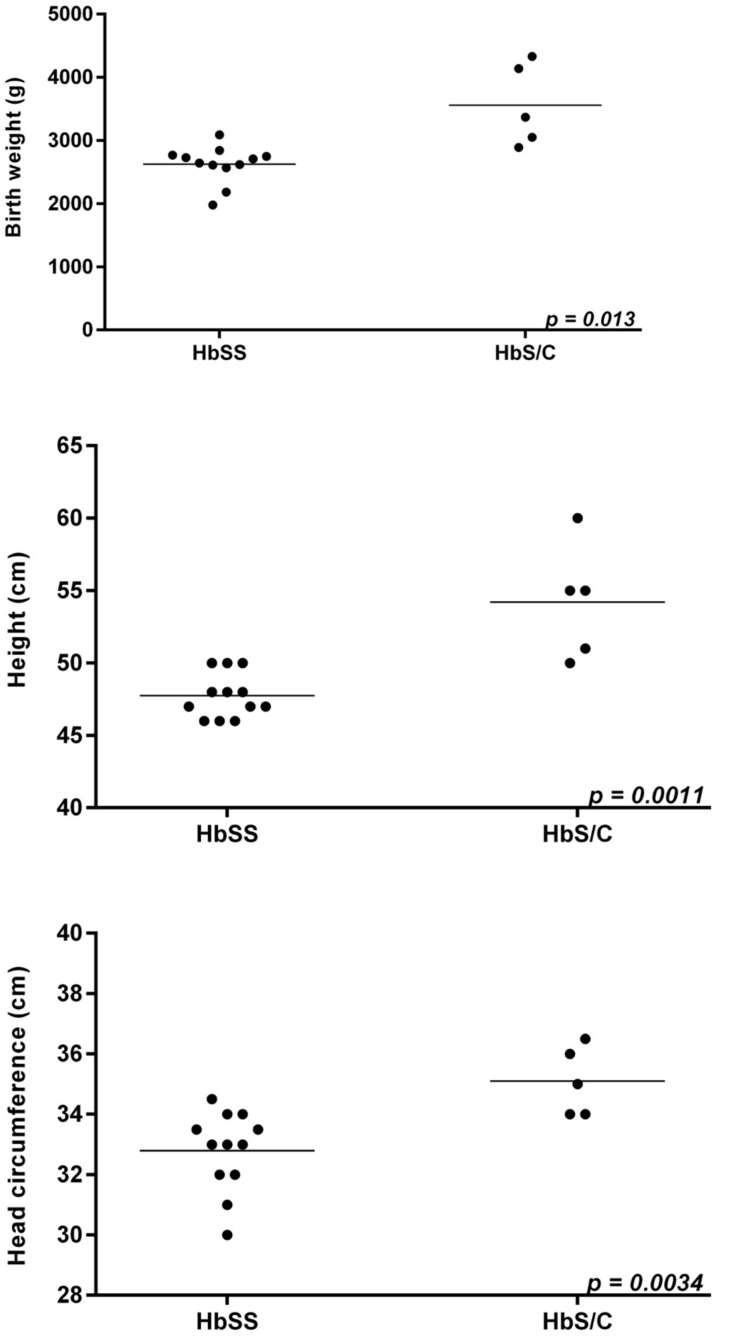
Neonatal status (Birth weight (g), height (cm), and head circumference (cm)) from women with HbSS (N = 12) versus HbS/C disease (N = 5) ±co-inheritance of heterozygous α-thalassemia, excluding preterm neonates.

**Table 1 jpm-11-00870-t001:** Maternal characteristics, including antenatal, peri-, and postpartum complications in SCD patients (N = 22).

Patient-ID	Genotype ((Genetic) Origin)	Gravida (G), Para (P); Age (Years)	Before Conception	Gestation Period	Country (Abortion/Stillbirth/Delivery)	Peripartum Period	Postpartum Period
			SCD-Associated Complication (HC Dosage; Alloantibodies)	SCD-Associated Complications		SCD-Associated Complications	SCD-Associated Complications
* **Exclusively succesful pregnancy outcomes** *							
1	HbSS-α^+^-thalassemia (Nigeria)	G1, P0; 39	recurrent VOCs/APCs, intermittent transfusion demand for RBCs (hemolytic/symtpomatic anemia, VOCs) (HC: 8 mg/kg BW; alloantibodies: anti-K)	VOCs, transfusion demand for RBCs (hemolytic/symptomatic anemia)	Germany	-	-
2	HbSS (Togo)	G1, P0; 32	recurrent APCs	APCs, transfusion demand for RBCs (hemolytic/symptomatic anemia; 1 unit)	Germany	CS secondary to birth arrest in the active-phase arrest	Transfusion demand for RBCs (2 units) due to symptomatic anemia (Hb 5.6 g/dL); Wound infection of the cesarean scar
		G2, P1; 38	recurrent APCs, st. p. cholecystectomy (cholecystolithiasis)	APCs	Germany	APCs	-
3	HbSS (Guinea)	G1, P0; 14	-	VOCs/APCs	Guinea	NDA	-
		G2, P1; 20	APCs (childhood), st. p. left hip joint endoprosthesis (left femoral head necrosis), st. p. cholecystectomy (cholecystolithiasis); HBV/HDV infection	VOCs, requiring inpatient admissions, UTIs	Germany	PROM, AIS	-
4	HbSS (Turkey)	G1, P0; 24	recurrent VOCs/APCs; intermittent transfusion demand for RBCs (hemolytic/symtpomatic anemia/VOCs), right femoral head necrosis, st. p. cholecystectomy (cholecystolithiasis)	VOCs, including ACS (RBC-exchange transfusion (GW 15)), requiring inpatient admissions, recurrent transfusion demand for RBCs (hemolytic/symptomatic anemia), UTIs	Germany	AIS (GW 37)	Transfusion demand for RBCs (2 units) (symptomatic anemia (Hb 6.5 g/dL))
		G2, P1; 27		VOCs, including ACS (RCE (GW 29+3)), requiring inpatient admissions, transfusion demand for RBCs (hemolytic/symptomatic anemia), UTIs	Germany	-	-
5	HbSS (Guinea)	G1, P0; 17 (Germany: GW 30; refugee status)	recurrent VOCs/APCs, intermittent transfusion demand for RBCs (hemolytic/symtpomatic anemia, VOCs (Guinea))	GW 30: VOC, requiring inpatient admission, transfusion demand for RBCs (hemolytic/symptomatic anemia), UTI	Germany	-	-
6	HbSS (Nigeria)	G1, P0; 26 (refugee status)	-	-	Germany	CS secondary to active-phase arrest, transfusion demand for RBCs (hemolytic/symptomatic anemia; 2 units)	-
		G2, P1; 28	splenic sequestration (RBC transfusion), cholecystolithiasis (HC 15 mg/kg BW)	VOCs/APCs, requiring inpatient admissions, transfusion demand for RBCs (hemolytic/symptomatic anemia), UTIs	Germany	transfusion demand for RBCs (PROM/hemolytic/symptomatic anemia; 2 units)	-
7	HbSS (Togo)	G1, P0; 17	recurrent VOCs/APCs, including ACS, intermittent transfusion demand for RBCs (hemolytic/symptomatic anemia, VOCs), cholecystolithiasis (27 mg/kg BW)	VOCs, requiring inpatient admissions (GW 9 and 27), transfusion demand for pRBCs (hemolytic/symptomatic anemia; 1 unit (GW 9)), UTI	Germany	Cervical insufficiency	-
8	HbS/β-thalassemia (Syria)	G1, P0; 33	recurrent VOCs/APCs, including ACS (pulmonary embolism); st. p. splenectomy (transfusion demand for pRBCs, hemolytic/symptomatic anemia, VOCs (Syria)), cholecystolithiasis (HC: 20 mg/kg BW)	Transfusion demand for RBCs (hemolytic/symptomatic anemia; Hb <7 g/dl; 5 units, GW 20, 25, 30 (borderline doppler indices; intrauterine growth restriction), APC	Germany	CS (secondary to pathologic CTG and doppler indices, placental insufficiency)	Transfusion demand for RBCs (2 units) due to symptomatic anemia (Hb 8.5 g/dL)
9	HbS/C (Ghana)	G1, P0; 30	-	-	Germany	-	-
		G2, P1; 31	-	-	Germany	-	-
		G3, P2; 32	-	-	Germany	AIS	-
10	HbS/C (Ghana)	G1, P0; 35	APCs	APCs	Germany	-	-
* **Successful and unsuccessful pregnancy outcomes** *							
11	HbSS-α^+^-thalassemia (Angola); Delta-storage pool disease (Diagnosis: 4th pregnancy)	G1, P0; 20	VOCs/APCs, intermittent transfusion demand for RBCs (hemolytic/symptomatic anemia, VOCs) (childhood), cholecystolithiasis, st. p. splenectomy	-	Germany (spontaneous abortion, GW 6)		
		G2, P0; 21		-	Germany (spontaneous abortion, GW 11)		
		G3, P0; 22		VOCs/APCs, transfusion demand for RBCs (1 unit)	Germany	VOC, transfusion demand for RBCs (hemolytic/symptomatic anemia; 1 unit)	-
		G4, P1; 25		VOCs/APCs, requiring inpatient admission (GW 26, 28), transfusion demand for RBCs (hemolytic/symptomatic anemia; 6 units); UTI-> urosepsis (GW 28)	Germany	CS (fetal subdural hematoma; GW 29 + 5); re-sectio (hematoma evacuation); re-re-sectio (post-bleeding); re-re-re-sectio (hysterectomy); re-re-re-re-sectio (hematoma evacuation); re-re-re-re-re-sectio (post-bleeding); hemorrhagic shock; DIC; ACS (pulmonary embolism); transfusion demand for RBCs (overall: 49 pRBC units) and platetes (16 units); postpartum cardiomyopathy (HFrEF; EF 20–35%)	
12	HbSS-α^+^-thalassemia (Nigeria)	G1, P0; 27	VOCs/APCs	APCs	Nigeria	-	-
		G2, P1; 30		APCs	Germany (elective abortion)		
		G2, P1; 33		APCs	Germany (spontaneous abortion)		
13	HbSS-α^+^-thalassemia (Zaire)	G1, P0; 17	VOCs/APCs, transfusion demand for RBCs (hemolytic symptomatic anemia, VOCs); st. p. splenectomy; st. p. cholecystectomy (cholecystolithiasis), infections (HC: 35 mg/kg BW)	-	Germany (elective abortion)		
		G2, P0; 19	VOCs/APCs, transfusion demand for RBCs (hemolytic symptomatic anemia, VOCs), left DVT (HC: 25 mg/kg BW)	VOCs/APCs, requiring inpatient admissions (GW 16, 32), transfusion demand for pRBCs (hemolytic/symptomatic anemia; 6 units), febrile UTIs	Germany	PROM, CS secondary to active-phase arrest	Transfusion demand for RBCs (hemolytic/symptomatic anemia; 1 unit)
		G3, P1; 22	VOCs/APCs, including ACSs (3rd ACS: vvECMO), transfusion demand for RBCs (hemolytic/symptomatic anemia, VOCs), gram-negative port infection (HC: 25 mg/kg BW)	VOCs/APCs, requiring inpatient admissions (GW 8, 11, 24, 29, 33), transfusion demand for RBCs (hemolytic/symptomatic anemia; 4 units)	Germany	PROM	-
14	HbSS (Dominican Republic)	G1, P0; 18	VOCs/APCs, including ACS, intermittent transfusion demand for RBCs (hemolytic/symptomatic anemia, VOCs), infections, cholecystolithiasis	VOCs/APCs, transfusion demand for RBCs (hemolytic/symptomatic anemia; 3 units)	Dominican Republic	-	-
		G2, P1; 21		-	Dominican Republic (spontaneous abortion)		
		G3, P1; 25		VOCs/APCs, transfusion demand for RBCs (hemolytic/symptomatic anemia; 2 units)	Dominican Republic	-	-
		G4, P2; 30		-	Germany (spontaneous abortion)		
15	HbSS (Congo)	G1, P0; 24	VOCs/APCs, including ACS; 4 yr. history of transfusion therapy (20–24th years of age; Italy)	VOCs/APCs, requiring inpatient admissions (SP(E)BT, every 3 weeks)	Italy	-	Continuation of transfusion therapy
		G2, P1; 29	VOCs/APCs (HC: 20 mg/kg BW)	VOCs, requiring inpatient admissions, transfusion demand for RBCs (hemolytic/symptomatic anemia; 3 units)	Germany	HELLP-Syndrome	-
		G3, P2; 32	VOCs/APCs (HC: 20 mg/kg BW)	-	Germany (spontaneous abortion)		
16	HbSS (Angola)	G1, P0; 20	VOCs/APCs, transfusion demand for RBCs (hemolytic/symptomatic anemia) (Angola/Belgium) (HC: 15 mg/kg BW; alloantibodies: anti- E, -M)	-	Belgium (sponatenous abortion)		
		G2, P1; 22 (Germany: GW 29, refugee status)		VOCs/APCs, requiring inpatient admissions (GW 29), transfusion demand for RBCs (hemolytic/symptomatic anemia; 2 units), possible DHTR (Anti-S) following RBC transfusion in the context of a VOC (DD febrile hyperhemolytic VOC (UTI)) (re-start HU (20 mg/kg BW), GW 34 + EPO)	Germany	vaginal bleedings (GW 35), PPROM	
		G3, P2; 24	VOCs/APCs, including ACS, transfusion demand for RBCs (hemolytic/symptomatic anemia) (HC: 20 mg/kg BW + EPO; alloantibodies: anti- E, -M, -S)	VOCs/APCs, requiring inpatient admissions (GW 13 and 17) (infectious-related), transfusion demand for RBCs (hemolytic/symptomatic anemia; 5 units) (EPO throughout pregnancy; re-start HU GW 30; 15 mg/kg BW)	Germany	-	-
17	HbS/C-α^+^-thalassemia (Ghana)	G1, P0; 17	VOCs/APCs	-	Germany (elective abortion)		
		G2, P0; 22		-	Germany	-	-
18	HbS/C (Nigeria)	G1, P0; 24	VOCs/APCs	-	Germany (elective abortion)		
		G2, P0; 26		APCs, transfusion demand for RBCs (hemolytic/symptomatic anemia; 2 units)	Germany	-	APCs
		G3, P1; 28		APCs	Germany	VOC-infectious related (UTI -> pyelonephritis) -> ACS	-
19	HbS/O-Ara (Kenya)	G1, P0; 28	recurrent VOCs (child- and adulthood), including recurrent ACSs (childhood) and transfusion demand for RBCs (symptomatic/hemolytic anemia, VOCs)	DVT	Germany (stillbirth, GW 26)		
		G2, P0; 28		-	Germany (spontaneous abortion, GW 11)		
		G3, P0; 29		VOCs	Germany	Placental abruption	-
* **Unsuccessful pregnancy outcomes** *							
20	HbSS-α^+^-thalassemia (Suriname)	G1, P0; 31	VOCs/APCs (childhood)	-	Germany (ectopic pregnancy (IVF); medical abortion)		
		G3, P0; 32	-	-	Germany (spontaneous abortion, (IVF))		
		G3, P0; 32	-	-	Germany (ectopic pregnancy (IVF); medical abortion)		
21	HbSS (Congo)	G1, P0; 22	Recurrent VOCs/APCs, including ACS, intermittent transfusion demand for RBCs (hemolytic/symptomatic anemia, VOCs; 8 units), right femoral head necrosis	-	Germany (elective abortion)		
		G2, P0; 24	recurrent VOCs/APCs, intermittent transfusion demand for RBCs (hemolytic/symptomatic anemia; 2 units), st. p. right hip joint endoprothesis (right femoral head necrosis), left humerus head necrosis (HC: 16 mg/kg BW)	-	Germany (elective abortion)		
		G3, P0; 28	recurrent VOCs/APCs, including ACS, intermittent transfusion demand for RBCs (hemolytic/symptomatic anemia; 2 units) (HC: 21 mg/kg BW)	-	Germany (elective abortion)		
		G4, P0; 28	recurrent VOCs/APCs (HC: 25 mg/kg BW)	-	Germany (spontaneous abortion)		
22	HbS-β-thalassemia (Cuba)	G1, P0; 18	recurrent VOCs/APCs, including ACS, st. p. splenectomy (splenic sequestration), transfusion demand for RBCs (hemolytic/symptomatic anemia, VOCs)	-	Germany (elective abortion)		
		G2, P0; 39	recurrent VOCs/APCs, including ACS, bi-femoral head necrosis (st. p. femoral head cannulation)	-	Germany (elective abortion)		

**Abbreviations:** ACS, acute chest syndrome; AIS, amniotic infection syndrome; APCs, acute pain crises; BW, body weight; CS, caesarean section; DHTR, delayed hemolytic transfusion reaction; DIC, disseminated intravascular coagulation; DVT, deep venous thrombosis; EPO, erythropoietin; GW, gestational week; Hb, hemoglobin; HC, hydroxycarbamide; HELLP, hemolysis, elevated liver enzymes, low platelets; HFrEF, heart failure with reduced ejection fraction; NDA, no data available; P(P)ROM, preterm (premature) rupture of membranes; RBC, red blood cells; SP(E)BT, serial prophylactic (exchange) blood transfusions; UTIs, urinary tract infections; VOCs, vaso-occlusive crises.

**Table 2 jpm-11-00870-t002:** Delivery and neonatal status in women with SCD (N = 31).

Patient-ID/Genotype (Mother)	Pregnancy	Delivery			Newborn						
		Time ^a^	Type	Country (Birth)	Apgar Score ^b^	ApH	VpH	BE (mmol/l)	Weight (g)	Height (cm)	Head Circumference (cm)
**HbSS (±α^+^-thalassemia)**											
1	1st	38 + 4	CS	Germany	9/10/10	7.25	7.3	−1.2	3090	50	34
2	1st	39 + 1	CS	Germany	8/10/10	7.16	-	−11.5	2620	48	33
	2nd	38 + 1	CS	Germany	9/10/10	7.25	-	−6.7	2730	50	33.5
3	1st	36	CS	Guinea	-	-	-	-	-	-	-
	2nd	37 + 1	VD	Germany	9/10 /10	7.2	-	-	2710	48	32
4	1st	37 + 5	CS	Germany	9/10/10	7.32	7.38	-	2610	47	34.5
	2nd	39 + 5	VD	Germany	9/10/10	7.29	7.36	-	2770	46	33.5
5	1st	37 + 5	VD	Germany	6/7/8	7.19	-	−10	1980	46	31
6	1st	38 + 3	CS	Germany	7/9/10	7.29	-	−5.8	2570	47	33
	2nd	35 + 1	CS	Germany	7/8/9	7.34	7.37	-	1900	40	31
7	1st	37 + 1	VD	Germany	9/10/10	7.10	7.27	11	2185	46	32
11	3rd	37 + 5	CS	Germany	9/10/10	7.33	7.37	0.4	2750	48	30
	4th	29 + 5	CS	Germany	7/8/8	7.37	-	-	1600	41	29
12	1st	36	CS	Nigeria	-	-	-	-	-	-	-
13	2nd	38 + 6	CS	Germany	9/10/10	7.32	7.37	−2.4	2845	47	33
	3rd	37 + 7	VD	Germany	9/10/10	7.29	7.31	-	2645	50	34
14	1st	39	VD	Dominican Republic	-	-	-	-	-	-	-
	3rd	40	CS	Dominican Republic	-	-	-	-	-	-	-
15	1st	32	CS	Italy	-	-	-	-	-	-	-
	2nd	34 + 1	CS	Germany	8/8/9	7.3	7.37		2320	-	-
16	2nd	35 + 2	VD	Germany	10/10/10	7.27	7.28	−5.6	2540	46	32
	3rd	34 + 4	VD	Germany	7/8/9	7.22	7.26	−4.5	2300	42	31
**HbS/β-thalassemia**											
8	1st	30 + 5	CS	Germany	-	7.21	-	-	946	36	27.2
**HbS/C**											
9	1st	39 + 0	VD	Germany	9/10/10	7.35	-	-	4140	55	36.5
	2nd	38	VD	Germany	-	-	-	-	-	-	-
	3rd	40 + 6	VD	Germany	9/10/10	7.29	-	−2.4	4330	55	36
10	1st	35 + 6	VD	Germany	9/10/10	7.38	-	-	2550	48	29.5
17	1st	40 + 0	VD	Germany	10/10/10	7.28	7.35	−3	3370	60	34
18	2nd	38 + 5	VD	Germany	9/10/10	7.29	-	-	3050	50	35
	3rd	37 + 3	CS	Germany	7/9/10	7.31	7.33	−2.6	2890	51	34
**HbS/O-Arab**											
19	3rd	29 + 2	CS	Germany	7/8/9	7.32			1140		

**Abbreviations:**^a^: Gestational week plus days; ^b^: After 1, 5, and 10 min; CS, caesarean section; VD, vaginal delivery; ApH, arterial pH; VpH, venous pH; BE, base excess.

## Data Availability

All datasets generated for this study are included in the article.

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
