# Peer review of "Pregnancies and Neonatal Outcomes in Patients with Sickle Cell Disease (SCD): Still a (High-)Risk Constellation?"

_jpm, 2021, doi:10.3390/jpm11090870_

Round 1

Reviewer 1 Report

The study carried out by the authors is an important study given the recent hightened attention to the sickle cell patient community - worldwide. The patients - being the most affected from a single point mutation - suffer a lifelong painful disorder with unpredictable crisis events and poor prognostic management. Pregnancy in sickle cell women are difficult due to the disease burden & unpredictable nature of the acute crises and other acute complications. Even though recent analysis of the registry on sickle patient population in Germany indicate a low prevalence in Germany - it is important that they receive proper care. The data presented in this study represents a snapshot of pregnancy burden in sickle patients and the pregnancy outcomes - which is necessary to understand the standard of care to facilitate any improvement required.

Author Response

Dear Reviewer,

thank you very much for your comments on our manuscript. An additional revision regarding grammar/spell check has been made, so we hope you agree with the revised version (Please see the attachment).

With best regards,

Ferras Alashkar

Reviewer 2 Report

This is a monocentric study of 25 years of observation to assess the outcome of maternal and fetal/neonatal outcome of the patients of SCD with 53 pregnancies. Screening was performed on 22 patients where 2 of the patients were diagnosed after pregnancy started. Patients were diagnosed for 96% with vaso-occlusive crisis, 13% with acute chest syndrome 79% with transfusion demand 42%, urinary tract infections, and 8% with thromboembolic events. Overall data supported for high rate of maternal and fetal/neonatal complications in pregnant SCD patients which support the results of previous studies.

1. Considering a small sample size, data supports the current existence of knowledge. Comparing the difference between pregnancy with well managed patients with SCD with late onset of treatment could be a good hypothesis to observe here. Earlier study found patients with an early diagnosis tend to have fewer vaso-occlusive crises. In this regards, it would be better to include the information regarding the duration of the treatment for SCD in these patients, to check if there is a difference with early onset of treatment with later onset.

2. The study includes 53 pregnancies but seven were not followed-up and the author did not explain directly if those patients were excluded from the study. It was mentioned that follow up was not possible because the patients moved out. If those patients were really excluded from the analysis, it will be better to add a sentence like: these patients were excluded from this study analysis.

3. Overall, the manuscript is well written, but some clarity is needed for well understanding of this study.

Author Response

Dear Reviewer,

thank you very much for your comments and careful reading.

In the following, we give a point-by-point reply to your comments:

  1. Considering a small sample size, data supports the current existence of knowledge. Comparing the difference between pregnancy with well managed patients with SCD with late onset of treatment could be a good hypothesis to observe here. Earlier study found patients with an early diagnosis tend to have fewer vaso-occlusive crises. In this regards, it would be better to include the information regarding the duration of the treatment for SCD in these patients, to check if there is a difference with early onset of treatment with later onset.

Answer: Thank you for this important advice. As a high rate of vaso-occlusive crises was observed in our patients and the patient survey revealed that not all patients seeked for advice of their treating physician during a painful crisis an answer to this important question cannot be adequately given. However, it was found that routine care was associated with a feeling of safety to contact the doctor directly in case of impending complications. Therefore, an additional sentence was added.

Added (discussion section):

Nevertheless, routine counselling was associated with a sense of security, according to the patient survey, as these patients could directly contact their attending physician in the event of impending complications, if desired. 

  1. The study includes 53 pregnancies but seven were not followed-up and the author did not explain directly if those patients were excluded from the study. It was mentioned that follow up was not possible because the patients moved out. If those patients were really excluded from the analysis, it will be better to add a sentence like: these patients were excluded from this study analysis.

Answer: All of the patients listed in the manuscript are registered at our center and continue to receive care through our clinic, so all patients continue to be followed up. To prevent any misunderstanding, the sentence below (results section, first paragraph) has been revised accordingly.  

Revised sentence: Prior to first presentation, seven gravidities occurred outside Germany, of which five were first-time pregnancies.

  1. Overall, the manuscript is well written, but some clarity is needed for well understanding of this study.

Answer: Following revision, we hope you agree with our revised version (Please see the attachment).

With best regards,

Ferras Alashkar